# CATM: A Multi-Feature-Based Cross-Scale Attentional Convolutional EEG Emotion Recognition Model

**DOI:** 10.3390/s24154837

**Published:** 2024-07-25

**Authors:** Hongde Yu, Xin Xiong, Jianhua Zhou, Ren Qian, Kaiwen Sha

**Affiliations:** Faculty of Information Engineering and Automation, Kunming University of Science and Technology, Kunming 650500, China; hmgdd@stu.kust.edu.cn (H.Y.); xiongxin840826@163.com (X.X.); qianren@stu.kust.edu.cn (R.Q.); shakaiwen@stu.kust.edu.cn (K.S.)

**Keywords:** emotion recognition, EEG, multi-feature, cross-scale attentional convolution

## Abstract

Aiming at the problem that existing emotion recognition methods fail to make full use of the information in the time, frequency, and spatial domains in the EEG signals, which leads to the low accuracy of EEG emotion classification, this paper proposes a multi-feature, multi-frequency band-based cross-scale attention convolutional model (CATM). The model is mainly composed of a cross-scale attention module, a frequency–space attention module, a feature transition module, a temporal feature extraction module, and a depth classification module. First, the cross-scale attentional convolution module extracts spatial features at different scales for the preprocessed EEG signals; then, the frequency–space attention module assigns higher weights to important channels and spatial locations; next, the temporal feature extraction module extracts temporal features of the EEG signals; and, finally, the depth classification module categorizes the EEG signals into emotions. We evaluated the proposed method on the DEAP dataset with accuracies of 99.70% and 99.74% in the valence and arousal binary classification experiments, respectively; the accuracy in the valence–arousal four-classification experiment was 97.27%. In addition, considering the application of fewer channels, we also conducted 5-channel experiments, and the binary classification accuracies of valence and arousal were 97.96% and 98.11%, respectively. The valence–arousal four-classification accuracy was 92.86%. The experimental results show that the method proposed in this paper exhibits better results compared to other recent methods, and also achieves better results in few-channel experiments.

## 1. Introduction

Emotion recognition has been applied in various fields, including mental health, medical diagnosis, intelligent driving, and distance education [1,2,3]. Early methods primarily relied on non-physiological signals such as voice intonation [4], facial expressions [5], and body movements [6]. However, these approaches are limited by subjectivity and susceptibility to manipulation. To address these limitations, methods based on physiological electrical signals have been developed. These signals include EEG, EMG [7], eye tracking [8], and ECG [9]. Among these, EEG provides rich information across time, frequency, and spatial domains, allowing for a more objective reflection of emotional states while minimizing the influence of external artifacts. Consequently, EEG is currently the preferred physiological signal for emotion recognition.

Some previous studies on EEG emotion recognition have not fully utilized the information in the time, frequency, and spatial domains, limiting emotional classification effectiveness. For instance, Han et al. [10] processed the original EEG signals using techniques like de-baselining and sliding window slicing, employing convolution and LSTM for feature extraction, achieving 96.28% accuracy in binary classification on the DEAP dataset. However, this approach relies on original EEG signals and fails to deeply mine entropy and energy features, focusing only on temporal information while neglecting spatial data. Lin et al. [11] utilized one-dimensional convolution and graph convolution to classify EEG emotions, achieving an average accuracy of 91% on the DEAP dataset. This method captures emotional features within channels and considers spatial relationships, yet it overlooks the temporal information in EEG signals, presenting significant limitations. Chen et al. [12] extracted spatial connectivity information from EEG channels, introduced a domain adaptation method, and constructed four connectivity matrices based on time series, achieving 95.15% accuracy in valence’s dichotomous classification on the DEAP dataset. However, this method does not account for the positional information between channels or the frequency domain and nonlinearity features in EEG signals, indicating room for improvement. Han et al. [13] proposed a multi-scale emotion recognition method (MS-ERM) that spatially mapped EEG signals and extracted temporal information using TimesNet. Validated on the DEAP dataset, it achieved average accuracies of 91.31% for arousal and 90.45% for valence classification. However, this method focuses solely on differential entropy features, neglecting energy, nonlinearity, and other important aspects, which limits classification effectiveness. Wu et al. [14] developed a 1D-Densenet model to extract frequency band energy and sample entropy from EEG signals, combining them into a one-dimensional input. This approach achieved an average binary classification accuracy of 93.51% on the DEAP dataset. While it effectively utilizes frequency domain, energy, and entropy features, it overlooks the temporal and spatial information present in the EEG signals. Singh et al. [15] introduced a hybrid model combining one-dimensional convolutional neural networks (1DCNNs) and bidirectional long short-term memory (Bi-LSTM), achieving binary and quadratic classification accuracies of 91.31% and 88.19%, respectively, on the DEAP dataset. This method also uses one-dimensional vectors as inputs, but ignores spatial information.

Most EEG-based emotion recognition studies have utilized full electrode arrays covering the entire scalp. However, with advancements in technology, portable, miniaturized EEG devices with fewer electrodes are becoming more common [16]. Studies using fewer channels have not consistently achieved better classification results, indicating significant room for improvement [17]. For example, Mert et al. [18] applied empirical modal decomposition (EMD) and multivariate expansion (MEMD) to process EEG signals, achieving dichotomous classification accuracies of 75% for arousal and 72.87% for valence using 18 channels from the DEAP dataset. This method does not account for the temporal continuity or spatial information of EEG signals and lacks channel selection, failing to fully leverage the available data. Bazgir et al. [19] employed Discrete Wavelet Transform (DWT) to decompose EEG signals into four frequency bands and extract spectral features. Using data from 10 channels in the DEAP dataset, they achieved binary classification accuracies of 91.3% for arousal and 91.1% for valence. However, this approach also did not extract the temporal and spatial features from the EEG signals.

To address these issues, we propose a multi-feature, multi-band spatio-temporal fusion model based on CATM. CATM performs feature extraction through cross-scale convolution, applies weight assignment using an attention mechanism, and ultimately classifies with a deep classifier. We conducted extensive binary classification, four-class classification, and experiments with fewer channels on the DEAP [20] public dataset, comparing our results with other models. The experimental findings demonstrate that our model exhibits excellent accuracy performance. The main contributions of this paper are as follows:In this study, we extracted four features at different frequencies: differential entropy, power spectral density, nonlinear energy, and fractal dimension. We spatially mapped these features according to the positions of the electrode channels, resulting in four-dimensional spatial features with enhanced discriminative and characterization capabilities. Finally, we performed feature fusion of multiple spatial features to leverage the advantages of each feature type.We propose a new spatio-temporal feature attention network to address the limitations of existing EEG emotion recognition methods. The network comprises a cross-scale attention convolution module, a transition module, a Bi-LSTM module, and a classifier. This architecture effectively enhances feature extraction capabilities and fully leverages the spatio-temporal features in EEG signals.To enable the proposed network model to fully utilize the information in the four-dimensional structure, we designed a frequency–space attention mechanism. This mechanism comprehensively considers the weights between convolutional channels and the positional relationships between electrode channels. It is embedded within the cross-scale convolutional module, allowing for adaptive weight assignment for both frequency and electrode channel positions in the EEG signal.Extensive experiments on the DEAP dataset demonstrate that the model exhibits strong emotion recognition accuracy and robustness in binary, four-class, and few-channel scenarios, validating the effectiveness of the proposed method.

## 2. Related Work

This section reviews the current research status of EEG emotion recognition, focusing on multi-feature approaches, machine learning, deep learning, attention mechanisms, and channel selection techniques.

### 2.1. Multi-Feature-Based EEG Emotion Recognition

EEG signals reflect the electrophysiological activity of brain neurons in the cerebral cortex or on the scalp surface. They offer advantages such as non-invasiveness, high temporal resolution, practicality, and cost-effectiveness, making them widely used in emotion recognition and motor imagery [21]. Extracting representative features from raw EEG data is crucial for subsequent classification, recognition, and analysis tasks. Common feature extraction methods include time domain analysis, frequency domain analysis, time–frequency domain analysis, multivariate statistical analysis, and nonlinear dynamic analysis [22,23,24]. Researchers typically categorize EEG signals into five frequency bands: delta, theta, alpha, beta, and gamma. Delta waves (0–3 Hz) are slow and associated with deep sleep, primarily observed in infants; they are not standard components in the EEG data of awake adolescents and adults, so this study focuses on four bands [25]. Theta waves (3–8 Hz) occur during relaxation or sleep. Alpha waves (8–14 Hz) are linked to visual processing, cognitive load, and memory activities. Beta waves (14–31 Hz) are prominent during conscious states such as calculation, reading, and thinking. Gamma waves (31–45 Hz) are associated with higher brain functions and are important for learning and memory. Singh et al. [26] explored single-dimensional and multidimensional EEG signal processing and feature extraction techniques across time, frequency, decomposition, time–frequency, and spatial domains. Yuvaraj et al. [27] extracted statistical features, fractal features, Hjorth parameters, higher-order spectral features, and wavelet coefficients from the DEAP dataset, classifying them with a shallow classifier, achieving valence and arousal accuracies of 78.18% and 79.90%, respectively. Liu et al. [28] proposed a dynamic differential entropy feature extraction algorithm that combines differential entropy (DE) with empirical modal decomposition (EMD). The resulting dynamic differential entropy features were classified using a convolutional neural network, yielding promising results. Çelebi et al. [29] utilized empirical wavelet transform (EWT) to decompose signals and extract frequency, linear, and nonlinear features from EEG data. They constructed a three-dimensional image and employed a deep learning framework with the DEAP dataset, achieving classification accuracies of 90.57% for valence and 90.59% for arousal in an across-subjects experiment. These studies indicate that extracting diverse EEG features and employing appropriate feature fusion methods can enhance emotion recognition compared to using single features [30].

### 2.2. Machine Learning-Based EEG Emotion Recognition

In early machine learning-based EEG emotion recognition, manual feature extraction was common, with these features applied to shallow classifiers. Currently, the mainstream approach involves automatic feature extraction through various algorithms, including time-domain, frequency domain, and time–frequency domain analyses. Extracted features are then applied to machine learning algorithms like Support Vector Machines (SVMs), k-Nearest Neighbor (KNN), and Naive Bayes (NB), achieving notable classification results [31]. Nawaz et al. [32] compared power, entropy, fractal metrics, statistical features, and wavelet features, using a feature selection algorithm, Principal Component Analysis (PCA), to attain validity and arousal accuracies of 77.62% and 78.96%, respectively. Bhardwaj et al. [33] classified seven different emotions using EEG signals; after preprocessing with filtering and independent component analysis (ICA), they extracted energy and power spectral density (PSD) features, achieving average classification accuracies of 74.13% with SVMs and 66.50% with Linear Discriminant Analysis (LDA). Gupta et al. [34] proposed a flexible analytic wavelet transform (FAWT) that decomposes EEG signals into different sub-band signals for feature extraction and smoothing. They classified the data using SVMs and Random Forests (RFs) on the SEED dataset, achieving an accuracy of 83.3%. Asghar et al. [35] introduced a deep neural network (DNN) method for EEG emotion recognition, extracting raw features from 2D spectrograms of each channel and employing dimensionality reduction with a pre-trained AlexNet model. They used SVM and k-NN classifiers on the SEED and DEAP datasets, achieving accuracies of 93.8% and 77.4%, respectively. These studies indicate that machine learning has made significant progress in emotion recognition based on EEG signals.

### 2.3. Deep Learning-Based EEG Emotion Recognition

With the advancement of deep learning, significant achievements have been made in various fields, including image processing and natural language processing. Recently, deep learning has also been widely applied to EEG emotion recognition. Li et al. [36] developed a convolutional recurrent neural network that combines convolutional and recurrent neural networks (LSTM), achieving binary classification accuracies of 72.06% for utility and 74.12% for arousal on the DEAP dataset. However, while the recurrent neural network captures temporal features, it does not effectively utilize spatial information. Chakravarthi et al. [37] introduced an automated CNN-LSTM model with Res-Net-152, achieving an impressive emotion recognition accuracy of 98% for human behavior and post-traumatic stress disorder (PTSD). Yang et al. [38] addressed the baseline signal’s impact on classification by removing it and combining channel position and frequency domain features to represent EEG signals as a three-dimensional structure, preserving spatial information. In their dichotomization experiments on the DEAP dataset, they achieved accuracies of 90.24% for validity and 89.45% for arousal.

Liu et al. [39] proposed the three-dimensional convolutional attentional neural network (3DCANN), which fuses spatio-temporal features with dual attention learning weights and uses a softmax classifier for emotion classification, achieving 97.35% accuracy on the SEED dataset. An et al. [40] constructed a spatio-temporal convolutional attention network, BiTCAN, which creates a two-dimensional mapping matrix of EEG signals based on electrode positions. This model extracts salient brain features using a bi-hemispheric disparity module and captures spatio-temporal features through a three-dimensional convolutional module. Extensive validation on the DEAP and SEED datasets resulted in accuracies exceeding 97% on both. Comparing these studies, it is evident that deep learning methods outperform traditional machine learning approaches in EEG-based emotion recognition, demonstrating superior feature extraction capabilities from EEG signals.

### 2.4. Attentional Mechanisms in EEG Emotion Recognition

The attention mechanism, rooted in the study of the human visual system, is a vital cognitive function. Researchers have adapted this concept for deep learning, integrating attention with neural networks. This integration allows neural networks to effectively receive target information, filter out irrelevant data, and allocate resources rationally. The core idea of the attention mechanism is to assign dynamic attentional weights that adjust during the learning process, enhancing the performance of the network by weighting raw data appropriately. Xiao et al. [41] introduced a four-dimensional attentional neural network that employs spectral and spatial attention mechanisms to assign weights adaptively across different brain regions and frequency bands, achieving 96.1% accuracy on the SEED dataset. Similarly, Jia et al. [42] developed a spatial–spectral–temporal attention 3D dense network, which attained accuracies of 96.02% and 84.92% on the SEED and SEED-IV datasets, respectively. The attention mechanism is crucial for optimizing deep learning models, allowing neural networks to focus on significant information during data processing, ultimately enhancing model performance.

### 2.5. Channel Selection in EEG Emotion Recognition

Selecting EEG channels that are highly relevant to emotions can significantly reduce the number of channels needed, making EEG signal collection more practical in daily life. This has become a key area of research. For instance, Zhang et al. [43] employed a ReliefF-based channel selection method on the DEAP dataset, achieving an optimal classification accuracy of 59.13% with 19 channels. Özerdem et al. [44] utilized Discrete Wavelet Transform (DWT) for feature extraction, implementing dynamic channel selection to identify five channels most relevant to emotions, resulting in a highest binary classification accuracy of 77.14%. Additionally, Topic et al. [45] applied ReliefF and Neighborhood Component Analysis (NCA) for channel selection, creating a holographic feature map. They achieved a maximum binary classification accuracy of 88.58% using data from just 10 channels in the DEAP dataset. These studies highlight the importance of effective channel selection in enhancing emotion recognition accuracy.

## 3. Materials and Methods

Figure 1 depicts the general framework and flow of the proposed method in this study. The EEG-based emotion classification method is divided into three parts: the first part is preprocessing and feature extraction, the second part is feature mapping and feature fusion, and the third part is feature extraction and classification using CATM. CATM mainly consists of five parts: the cross-scale attention module (CSAM), frequency–space attention module (FSAM), feature transition module (FTM), temporal feature extraction module (Bi_LSTM), and deep classification module (DCM). Table 1 shows the acronyms, full names, and functions of each CATM module. The next sections describe the model in detail and evaluate the model.

### 3.1. Feature Extraction

The EEG signal is characterized by a low signal-to-noise ratio, a significant presence of low-frequency components, and a waveform that displays changing nonlinear characteristics. Building on these traits, this study aims to synthesize frequency domain, time domain, and nonlinear features from EEG data to enhance emotion recognition accuracy. First, for each subject, the raw EEG data are divided into N non-overlapping, equal-length segments, with labels assigned to each segment. Each segment is then decomposed into four frequency bands, theta (3–8 Hz), alpha (8–14 Hz), beta (14–31 Hz), and gamma (31–45 Hz), using a Butterworth filter. Subsequently, data from each band are extracted using four specific features: differential entropy (DE), power spectral density (PSD), nonlinear energy (NE), and fractal dimension (FD). The feature extraction and mapping process is illustrated in Figure 2.

Differential entropy (*DE*) features are widely used by researchers for emotion recognition in EEG signals and have been shown to be the most stable features [46]. The *DE* of the EEG signal is an extension of the Shannon entropy on continuous variables, which allows us to distinguish the low- and high-frequency energies of the EEG signal and is calculated as follows:(1)$$DE=-\int_{a}^{b}\frac{1}{\sqrt{2\pi \sigma_{i}^{2}}}e^{-\frac{{\left(x-\mu \right)}^{2}}{2\sigma_{i}^{2}}}\log(\frac{1}{\sqrt{2\pi \sigma_{i}^{2}}}e^{-\frac{{\left(x-\mu \right)}^{2}}{2\sigma_{i}^{2}}})dx=\frac{1}{2}\log(2\pi e\sigma_{i}^{2})$$
where e represents Euler’s constant and σ represents the standard deviation of x. The *DE* is calculated for an EEG signal of length [a, b] and approximately obeys a Gaussian distribution $N(\mu ,\sigma_{i}^{2})$, which is equal to the logarithm of the energy spectrum in a particular frequency band.

Power spectral density (PSD) features have been shown to be effective in *EEG* emotion recognition in previous studies [47], and in this study, the average power of a segment of an EEG signal is described. If there is a segment of *EEG* signal of length M denoted as x(t), and the value of t is taken as 0~M−1, the PSD formula for this segment is as follows:(2)$$P(\omega_{k})=\sum_{t=-(M-1)}^{M-1}\gamma (t)e^{-j\omega_{k}t}$$
where $P(\omega_{k})$ is the power spectral density, γ(t) is the autocorrelation function of x(t), $\omega_{k}$ is the angular frequency, t is the time, and γ(t) is the autocorrelation function of x(t). γ(t) is calculated as follows:(3)$$\gamma (t)=E\left[x(n)x^{*}(n+t)\right]$$
where E is the expectation of the function and $x^{*}$ is the complex conjugate of x.

Nonlinear energy (*NE*) was proposed by Toole et al. [48] and is widely used in areas such as epileptogenesis detection by EEGs [49]. Nonlinear energy takes into account the product of amplitude squared and frequency squared and is used to calculate the instantaneous energy of a signal, especially for identifying transient changes. The first-order difference of the signal and the Hilbert transform were utilized in this study to obtain an estimate of the nonlinear energy. The average *NE* of a segment of an *EEG* signal is calculated as follows:(4)$$NE=\frac{1}{N}\sum_{n=1}^{N}\Gamma (x(n))$$
where Γ(x(n)) is the estimated value of the nonlinear energy. The formula Γ(x(n)) is given below:(5)$$\Gamma (x(n))={\left(y(n)\right)}^{2}+|y_{\mathrm{hilbert}}(n){|}^{2}$$
where y(n) denotes the first-order difference of the EEG signal and $y_{\mathrm{hilbert}}(n)$ denotes the Hilbert transform of y(n).

Fractal dimension (FD) [50] is a nonlinear feature used to quantify *EEG* signals, which captures the self-similarity and complexity of the signal, and can characterize the temporal structure and dynamics of the signal. It is widely used in the fields of disease diagnosis from EEG signals [51] and sleep monitoring [52]. *FD* features have also been applied to research related to EEG emotion recognition [53], which has achieved some results in few-channel emotion recognition and demonstrated that classification accuracy can be improved by using *FD* features. *FD* features are generally calculated by measuring the self-similarity of the signal; due to the computational difficulties in estimating the fractal dimension of complex signals, this paper adopts the Petrosian fractal dimension approximation, which is calculated as follows:(6)$$FD=\frac{\log_{10}(N_{EEG})}{\log_{10}(\frac{N_{EEG}}{\left(N_{EEG}+0.4N_{\Delta }\right)})}$$
where $N_{EEG}$ is the number of sampling points of the *EEG* signal and $N_{\Delta }$ is the number of sign changes in the signal.

In this study, four features will be extracted for each frequency band within the 0.5 s time window. Due to the presence of low-frequency noise in the *EEG* baseline signal, in order to effectively avoid noise effects and baseline drift, the baseline correction was performed by subtracting the average features of the first 3 s baseline signal from the features of the last 60 s, and finally, all the data were normalized to obtain the one-dimensional *EEG* features.

### 3.2. Feature Mapping and Feature Fusion

EEG signals also contain crucial spatial location information. To fully utilize the temporal, spatial, and frequency aspects of these signals, we constructed four-dimensional structural features to describe them. As illustrated in Figure 3, we mapped the features from the 32-channel EEG signals into a two-dimensional feature matrix based on the 32-electrode layout of the international 10–20 system. Positions without corresponding channels were filled with zeros, resulting in a two-dimensional feature map sized 8 × 9. The features from each frequency band of the EEG signals were mapped to these 2D feature maps. Subsequently, the 2D feature maps for the four frequency bands were stacked to create 8 × 9 × 4 3D features. This approach allows for a comprehensive representation of the EEG data, integrating multiple dimensions of information.

The most commonly used feature fusion methods in deep learning-based algorithms are feature matrix addition, multiplication, and splicing [54]. According to a previous study [55], the use of feature matrix splicing is better than that of matrix addition and multiplication, so in this study, the feature splicing method is used to realize the specific formula as follows:(7)$$X_{DE}=\left[d_{1},d_{2},d_{3},\cdots ,d_{N}\right]$$
(8)$$X_{PSD}=\left[p_{1},p_{2},p_{3},\cdots ,p_{N}\right]$$
(9)$$X_{NE}=\left[n_{1},n_{2},n_{3},\cdots ,n_{N}\right]$$
(10)$$X_{FD}=\left[f_{1},f_{2},f_{3},\cdots ,f_{N}\right]$$
(11)$$X_{con}=\left[X_{DE};X_{PSD};X_{NE};X_{FD}\right]=\left[d_{1};p_{1};n_{1};f_{1},\cdots ,d_{N};p_{N};n_{N};f_{N}\right]$$

In Equations (7)–(11), d denotes the three-dimensional features of *DE*, whose four-dimensional features are denoted by $X_{DE}$; p denotes the three-dimensional features of PSD, whose four-dimensional features are denoted by $X_{PSD}$; n denotes the three-dimensional features of *NE*, whose four-dimensional features are denoted by $X_{NE}$; and f denotes the three-dimensional features of *FD*, whose four-dimensional features are denoted by $X_{FD}$; these features are fused to obtain the feature set $X_{con}$. Since the EEG signal of each subject was divided into *N* equal-length segments according to time, a four-dimensional feature of N × 16 × 8 × 9 was obtained. The feature fully preserves the temporal, spatial, and frequency information in the EEG signal, laying the foundation for subsequent feature extraction and classification.

### 3.3. Network Model Architecture and Classification

#### 3.3.1. Cross-Scale Attention Module (CSAM)

CSAM captures features at different scales by using convolutional kernels of different sizes in parallel and combining them with a frequency–space attention mechanism. Although CSAM is sparsely structured, it produces dense feature data. The CSAM module uses three different sizes of convolutional kernels (1 × 1, 3 × 3, 5 × 5) and a maximum pooling kernel (3 × 3). Convolutional kernels of different sizes have different receptive fields, and the use of different sizes of convolution can extract subtle *EEG* features and take into account the relative positional relationships between individual electrodes. The spatial dimensions of the feature maps are compressed using the maximum pooling layer to further extract more abstract and advanced features by reducing the width and height of the feature maps. Adding the BN layer and ReLU activation function after the convolution and pooling operations effectively prevents overfitting and increases the nonlinear characteristics of the network. Finally, the CSAM module is obtained after adding the frequency–space attention mechanism to the ReLU activation function. The structure of the CSAM module is shown in Figure 4.

The parameter settings of the CSAM module are shown in Table 2.

#### 3.3.2. Frequency–Space Attention Module (FSAM)

It has been shown [56] that different frequency bands of *EEG* signals have different recognition effects in emotion recognition. The β and γ bands have better recognition performance, followed by the α band, while the θ band is the least effective. When mapping 2D feature maps, there are regions where electrode positions are missing and we use zeros to fill in these positions, but this introduces a lot of useless information. To enable the network model to better extract useful information from the feature maps, we introduced a frequency–space integrated attention mechanism to give higher weights to the more important frequency bands and spatial locations to better utilize the *EEG* information related to emotions. The structure of the frequency–space attention mechanism network is shown in Figure 5.

The input tensor in Figure 5 is B × C × H × W. B, C, H, and W denote the batch size, number of channels, height, and width of the tensor, respectively. For a given input feature map $X\in R^{B\times C\times H\times W}$, the feature map $Y\in R^{B\times C\times H\times W}$ is obtained after going through the frequency attention mechanism $F_{f}$ and the spatial attention mechanism $F_{s}$ in turn. This is expressed as the following equation:(12)$$X^{\prime }=F_{f}(X)⊗X$$
(13)$$Y=F_{s}(X^{\prime })⊗X^{\prime }$$
where $X^{\prime }$ in Equations (12) and (13) denotes the output of the feature map after going through the frequency attention mechanism $F_{f}$.

The frequency attention mechanism mainly acts on different channels of the feature map and can effectively label important frequency bands in the *EEG* signal. First, a global adaptive average pooling operation on the spatial dimension of $X\in R^{B\times C\times H\times W}$ is performed to obtain $X_{avg}\in R^{B\times C\times 1\times 1}$, which is computed as follows:(14)$$X_{avg}=\mathrm{Adaptive}\_\mathrm{AvgPool}(X(h,w))$$

$X_{avg}$ is then fed into the multilayer perceptron (MLP), which consists of two fully connected layers and *ReLu* and Sigmoid activation functions. The $F_{ReLu}(x)$ and $F_{Sigmoid}(x)$ activation functions are denoted as
(15)$$F_{ReLu}(x)=\left\{\begin{array}{c} x\;(x>0)\\ 0\;(x\leq 0) \end{array}\right.$$
(16)$$F_{Signod}(x)=\frac{1}{1+e^{-x}}$$

Finally, the frequency attention mechanism $F_{f}\in R^{B\times C}+b$ is realized by going through the linear layer and adding the bias term b. By applying the frequency attention weights of the output to the input signal, different weights can be assigned to the individual channels, and emotionally relevant frequency bands will be given more attention. As The different weights assigned by the frequency attention mechanism to the 464 channels in CSAM are shown in Figure 6.

The spatial attention mechanism mainly acts on different spatial locations of the feature map, and for the feature $X^{\prime }$ output from the frequency attention mechanism, ${X^{\prime }}_{\max}$ is first obtained through the maximum pooling layer:(17)$${X^{\prime }}_{\max}=Max\mathrm{Pool}(X^{\prime }(h,w))$$

After that, the spatial attention mechanism $F_{s}\in R^{B\times 1\times H\times W}$ is obtained by two-dimensional convolution and *ReLu* and Sigmoid activation functions, and then the final output Y of the frequency–space attention mechanism can be obtained by multiplying $F_{s}$ with the feature $X^{\prime }$. Figure 7 represents the 3 epochs for random training of the model, which are features in CSAM. After going through the spatial attention mechanism, different electrode positions were given different weights for the thermogram. It is easy to see that the frontal part of the brain is weighted higher and the parietal part is weighted lower.

#### 3.3.3. Feature Transition Module (FTM)

Multiple features of *EEG* signals were used, and feature fusion was performed in this study. Feature splicing was also used to fuse feature maps of different sizes in the subsequent cross-scale convolution, with more channels of the output feature maps, all of these operations doubling the dimensionality of the input feature maps and occupying more memory. In order to control memory consumption while maintaining network performance, transition modules are introduced. The transition module normalizes the input feature map by a batch normalization layer, which enables obtaining more emotionally expressive and stable low-dimensional features. The network structure of the feature transition module is shown in Figure 8, where the size of the maximum pooling kernel is set to 3.

#### 3.3.4. Temporal Feature Extraction Module (Bi-LSTM)

The bidirectional long short-term memory (Bi-LSTM) network can process temporal data efficiently by using two LSTM units at each time step, one for processing sequences from front to back and the other for processing sequences from back to front. The final output is obtained by superimposing the results of LSTM computations in two different directions. The LSTM cell consists of an input gate, an oblivion gate, and an output gate, which can be defined as
(18)$$i_{t}=\sigma (W_{i}\cdot \left[h_{t-1},q_{t}\right]+b_{i})$$
(19)$$f_{t}=\sigma (W_{f}\cdot \left[h_{t-1},q_{t}\right]+b_{f})$$
(20)$$g_{t}=\tan h(W_{c}\cdot \left[h_{t-1},q_{t}\right]+b_{c})$$
(21)$$C_{t}=f_{t}C_{t-1}+i_{t}g_{t}$$
(22)$$o_{t}=\sigma (W_{o}\cdot \left[h_{t-1},q_{t}\right]+b_{o})$$
(23)$$h_{t}=o_{t}\tan h(C_{t})$$

In Equations (18)–(23), $\left[h_{t-1},q_{t}\right]$ denotes the current input and the previous state, W represents its corresponding weight matrix, σ is the Sigmoid function, and b is the bias term. The final output is obtained from the output of the forward LSTM and the reverse LSTM, denoted as $y_{t}=\left[h_{t};h_{t}^{\prime }\right]$.

#### 3.3.5. Deep Classification Module (DCM)

Since deep classifiers tend to be more effective than shallow ones, for the extracted temporal feature output, a deep classifier for emotion classification is designed in this study. The classifier consists of a fully connected layer, a ReLU activation function, and Dropout regularization. The deep classifier network structure is shown in Figure 9, and the specific parameters are shown in Table 3.

## 4. Experiments

### 4.1. Dataset and Dataset Processing

The DEAP dataset is a physiological signaling dataset generated using music videos as eliciting materials. During the experiment, each subject watched 40 min of music videos and assessed their emotions based on subjective feelings of arousal, valence, liking, and dominance. A total of 32 subjects participated, and each experimental session consisted of signals from 40 channels. The first 32 channels recorded EEG signals, while the last 8 channels included other physiological signals, such as ocular and EMG data. Each experiment lasted 63 s, with the first 3 s capturing baseline signals and the remaining 60 s reflecting emotion-evoked signals. The dataset’s sampling frequency was downsampled from 512 Hz to 128 Hz, resulting in a physiological signal matrix for each subject of size 40 × 40 × 8064 (40 experimental music clips, 40 physiological signal channels, and 8064 sampling points). The DEAP dataset offers rich data on EEG and other physiological signals, along with subjective emotion assessments from participants, providing valuable insights into the relationship between emotion and physiological responses.

Valence and arousal tags were selected in the DEAP dataset, and tags scoring above 5 were assigned a value of 1 and tags below 5 were assigned a value of 0. Dichotomous experiments can be categorized as high valence (HV) and low valence (LV) or high arousal (HA) and low arousal (LA). The four-classification experiment (V-A) can be categorized as high valence high arousal (HVHA), high valence low arousal (HVLA), low valence high arousal (LVHA), and low valence low arousal (LVLA). HVHA corresponds to a state of excitement or agitation, HVLA to a state of calmness or relaxation, LVHA to depression or anger, and LVLA to frustration or sadness.

### 4.2. Experiment Setup and Performance Evaluation Metrics

All experiments used the same software environment, experimental dataset division, parameter settings, and evaluation metrics. The software environment is the Windows 11 operating system, Python 3.9 programming language environment, and Pytorch deep learning framework, and the hardware uses NVIDIA GeForce RTX 4060 GPU (NVIDIA: Santa Clara, CA, USA). The model is trained using Adam as the optimizer, and cross-entropy and L2 regularization as the loss function, and the learning rate is set to 0.001. To reduce the overfitting phenomenon during training, the parameter of Dropout was set to 0.5. We employed ten-fold cross-validation. Each participant’s dataset was divided into ten parts, and in each experiment, one part was used as the test set while the remaining nine parts were used as the training set. This procedure was repeated ten times, and the average of the ten experiments was taken as the final result. Therefore, in each experiment, the training set and the test set were split in a ratio of 9:1. Finally, the average of the 32 subjects’ identifications was taken as the result of the dataset. To objectively evaluate the performance of the model, this study uses the accuracy, precision, recall, and *F*1-*score* as evaluation metrics, and the calculation formulas are as follows, respectively:(24)$$Accuracy=\frac{TP+TN}{TP+FP+TN+FN}$$
(25)$$precision=\frac{TP}{TP+FP}$$
(26)$$Recall=\frac{TP}{TP+FN}$$
(27)$$F1\_score=2\times \frac{precision\times Recall}{perccision+Recall}$$

In Equations (24)–(27), *TP* denotes that the true label is a positive class and is predicted to be positive, *TN* denotes that the true label is a negative class and is predicted to be negative, *FP* denotes that the true label is a negative class but is predicted to be positive, and *FN* denotes that the true label is a positive class but is predicted to be negative.

## 5. Results and Discussion

To verify the effectiveness of the proposed method, this study conducts ablation experiments and comparative tests on the DEAP dataset and evaluates the model performance with evaluation metrics. Figure 10 shows the accuracy of the 32 subjects in the DEAP dataset in the valence dimension, arousal dimension, and valence–arousal dimension. The accuracies of the proposed method in this study are 99.70% and 99.74% for the valence dimension and arousal dimension, respectively, and 97.27% for V-A. Figure 11 shows the confusion matrices of the CATM experiment on the DEAP dataset.

### 5.1. Ablation Experiments

In this study, feature fusion ablation experiments, feature ablation experiments, and module ablation experiments were conducted.

#### 5.1.1. Feature Fusion Ablation Experiments

The commonly used feature fusion methods in deep learning algorithms are feature addition, multiplication, and concatenation, and this experiment discusses the effect of different feature fusion strategies on the effect of emotion recognition. The results of the experiments are shown in Table 4.

In the table, $X_{add}$ means four 3D features are added together, $X_{mult}$ means four 3D features are multiplied together, and $X_{con}$ means four 3D features are concatenated together.

#### 5.1.2. Feature Ablation Experiments

The feature ablation experiments were performed on four features, DE, PSD, NE, and FD, respectively. The experimental results are shown in Table 5.

From the experimental results, it can be seen that the four features have the best classification accuracy compared to single, double, and triple features due to the fact that the introduction of multi-features helps to improve the accuracy of emotion recognition, and the experimental results also prove the effectiveness of PSD features, FD features, and NE features for emotion recognition.

#### 5.1.3. Module Ablation Experiments

CATM mainly consists of a cross-scale attention module, a frequency–space attention module, a spatio-temporal feature extraction module, and a deep classification module. We ablate each module to verify its contribution in the classification task. For a better representation, we use Model 1 to represent the model that lacks the cross-scale attention module, Model 2 to represent the model that lacks the frequency–space attention module, Model 3 to represent the model that lacks the temporal feature extraction module, and Model 4 to represent the model that lacks the deep classification module. Detailed information is shown in Table 6.

As can be seen from Table 7, after ablating the cross-scale attention module, the validity and arousal dimension classification accuracies decreased by 8.36% and 4.68%, respectively; the ablation of the frequency–space attention module resulted in decreases of 0.75% and 0.77% in potency and arousal dimension classification accuracies, respectively; after ablating the temporal feature extraction module, the classification accuracies of the potency and arousal dimensions decreased by 7.95% and 7.29%, respectively; and after ablating the deep classification module, the classification accuracies of the potency and arousal dimensions decreased by 0.78% and 0.87%, respectively. It can be seen that after ablating each module, the feature extraction ability of the model is reduced, and the temporal, spatial, and frequency information in the EEG signal is not fully utilized. Figure 12 demonstrates the accuracy of each subject in the valence and arousal dimensions in the model ablation experiment. We found that the recognition accuracy of individual subjects decreased after ablating certain modules. Module ablation experiments demonstrate the validity of individual modules in the model.

In order to verify the performance of the cross-scale attention module in the model, we replaced the module with a single-scale convolution with convolution kernel sizes of 1, 3, and 5 for the experiments, and the results are shown in Table 8.

Table 8 shows that there is a decrease in model performance after replacing CSAM with single-scale convolution. A possible reason is that single-scale convolution has a relatively limited capability for feature extraction, making it unable to capture multi-scale features.

### 5.2. Experiments with Few Channels

To demonstrate the validity of this method for use with fewer channels, we selected electrodes with a high correlation to emotion based on a previous study [12]. We conducted experiments using data from these channels in the DEAP dataset. Experiments were performed with both 5 and 18 electrodes. The 5 selected electrodes were Fp1, Fp2, F7, F8, and O1. The 18 electrodes included Fp1, Fp2, F7, F3, Fz, F4, F8, T7, C3, CZ, C4, T8, P7, P3, P4, P8, O1, and O2. The electrode mapping is shown in Figure 13.

The experimental results for the fewer channels are shown in Table 9 and Table 10.

According to Table 9 and Table 10, the method in this study demonstrates good performance in experiments with fewer channels. In the 5-channel experiments, the accuracies for valence and arousal were 97.96% and 98.11%, respectively, while the V-A classification achieved an accuracy of 92.86%. For the 18-channel experiment, the accuracies were 99.59% for valence and 99.53% for arousal, with the V-A classification achieving 94.57%. The accuracy for each subject is illustrated in Figure 14 and Figure 15.

According to Figure 14 and Figure 15, there was a noticeable decrease in the experimental categorization accuracy of individual subjects following the reduction in EEG channels. Specifically, in the 5-channel experiments, both dichotomous and quaternary classifications yielded lower accuracy for individual subjects compared to the 18- and 32-channel configurations.

In the 5-channel experiment, we selected EEG channels from the frontal and occipital regions. The literature [57] indicates that the prefrontal, parietal, and occipital areas may contain the most relevant information for emotion recognition, consistent with previous studies [58,59]. Additionally, references [60,61] noted that synchronization between the frontal and occipital lobes is associated with both positive and fearful emotions. This study conducted experiments using all channels (32), 18 channels, and 5 channels. The results showed that using more channels improved the model’s classification performance, aligning with previous research findings [62]. Reducing the number of channels decreases the input EEG data to the model, significantly lowering both the parameter count and computational cost. However, having fewer data may hinder the model’s ability to perform classification tasks, potentially affecting its performance. Our experiments demonstrated that CATM still achieved good classification results even with only 5 channels.

In recent years, portable EEG acquisition devices have gained popularity, offering options for 15 channels, 5 channels, and even 1 channel. Limited channels can capture only low-density EEG signals, and since emotions are represented across multiple brain regions, extracting effective emotional patterns from such limited data poses significant challenges.

### 5.3. Comparative Experiments

We compare the proposed sentiment classification model with recent sentiment classification models, as shown in Table 11.

The following is a characterization of the various methods in Table 11:(1)FSA-3D-CNN [63]: This method constructs a 3D matrix of the EEG containing spatio-temporal information and introduces an attention mechanism to use 3D-CNN for emotion classification tasks.(2)TSFFN [64]: This method performs de-baselining of the EEG and extracts spatio-temporal features from EEG signals using a parallel transformer and a three-dimensional convolutional neural network (3D-CNN), and finally performs an emotion classification task.(3)Multi-aCRNN [65]: This method proposes a multi-view feature fusion attentional convolutional recurrent neural network. The interference of label noise is reduced by label smoothing, and GRU and CNN are combined to accomplish the emotion classification task.(4)RA2-3DCNN [66]: This method introduces segmentation–transformation–merge techniques, residuals, and attention mechanisms into shallow networks to improve the accuracy of the model. It is based on the 2D convolutional neural network and 3D convolutional neural network for emotion recognition.(5)MDCNAResnet [67]: This method extracts differential entropy features from EEG signals and constructs a three-dimensional feature matrix, uses deformable convolution to extract high-level abstract features, and combines MDCNAResnet with bidirectional gated recurrent units (BiGRUs) to accomplish emotion recognition.(6)BiTCAN [40]: This method utilizes a bi-hemispheric difference module to extract salience features of brain cognition, fuses salience and spatio-temporal features into an attention module, and inputs them into a classifier for emotion recognition.(7)RFPN-S2D-CNN [68]: This method uses preprocessed signals, differential entropy (DE), symmetric difference, and the symmetric quotient to construct four EEG signal feature matrices, and proposes a residual feature pyramid network (RFPN) to obtain inter-channel correlation, which is effective in improving the classification accuracy of emotion recognition.(8)FCAN-XGBoost [55]: This method extracts DE features and PSD features of the EEG and fuses FCAN and XGBoost algorithms for sentiment recognition, which reduces computational cost and improves classification accuracy.(9)Multi-scale 3D-CRU [69]: This method reconstructs a 3D feature representation of the EEG containing delta (δ) frequencies, combined with a recurrent neural network GRU for emotion classification.(10)MES-CTNet [70]: This method proposes a new capsule transformer network based on multidomain features, which uses multiple features and multiple attention mechanisms, and achieves high accuracy in emotion classification.

The proposed method in this study utilizes various features of the EEG and combines a cross-scale attentional convolution module with a temporal feature extraction module to extract temporal–frequency–spatial features in EEG signals, enhancing the accuracy of emotion classification. The data in Table 11 indicate that our method outperforms other approaches in both dichotomous and quaternary tasks.

We obtained some information about the CATM through experiments and built-in functions in Pytorch. The model has a total of 9,478,504 parameters. Under the previously mentioned hardware conditions, it takes 293.31 ms to process the EEG data of one subject. The size of the model’s weight file is 36.07 MB. We also hope that other researchers will include model-related information in their studies to facilitate comparisons in future research.

In summary, the multi-feature, multi-band emotion recognition method proposed in this study offers significant advantages over other recent approaches. The effectiveness of various features and modules is validated through multiple ablation experiments. In tests with fewer channels, the method achieves high emotion classification accuracy using only 5-channel and 18-channel data. Additionally, the recognition accuracy is more balanced across subjects, and the performance is more stable compared to single-feature methods, making it better suited for real-world applications.

However, this method has some shortcomings. While it shows better performance in within-subject experiments, it performs poorly in cross-subject experiments. The reason for this may be that physiological representations and subjective feelings differ across individuals in terms of cross-subject emotion recognition. Cross-subject experiments will be favored in future research, more integrated with practical applications.

## 6. Conclusions

In this study, we propose a multi-feature, multi-band spatio-temporal fusion algorithm called CATM. First, we extract DE, PSD, NE, and FD features from the EEG signals across different channels. These features undergo de-baselining and are organized into a 3D feature matrix based on the relative positions of electrodes, fully utilizing the frequency domain, energy, nonlinearity, spatio-temporal complexity, and spatial information in the EEG signals. Next, a cross-scale attention module is employed to extract spatial features at different scales within the EEG signals. The extracted features receive varying weights at spatial locations and channels through a frequency–space attention mechanism, enhancing the classification performance of the model. To prevent overfitting and reduce computation, a transition module is introduced to improve model generalization. Finally, Bi-LSTM is utilized to extract the temporal features from the EEG signals, facilitating the fusion of spatio-temporal features.

The experimental results of the proposed method on the DEAP dataset demonstrate its effectiveness in extracting emotion-related features from EEG signals. The classification accuracies achieved were 99.70% for valence, 99.74% for arousal, and 97.27% for the combined valence–arousal dimension. In experiments using fewer channels, the 5-channel EEG signal yielded classification accuracies of 97.96% for valence and 98.11% for arousal, with a 92.86% accuracy for the combined dimension. In the 18-channel experiment, the accuracies were 99.59% for valence, 99.53% for arousal, and 94.57% for the combined dimension. In our future work, we aim to optimize the network structure to reduce parameters and computational costs while enhancing classification accuracy. Additionally, we plan to conduct experiments that combine multimodal data to mitigate the impact of individual differences on network models.

## Figures and Tables

**Figure 1 sensors-24-04837-f001:**
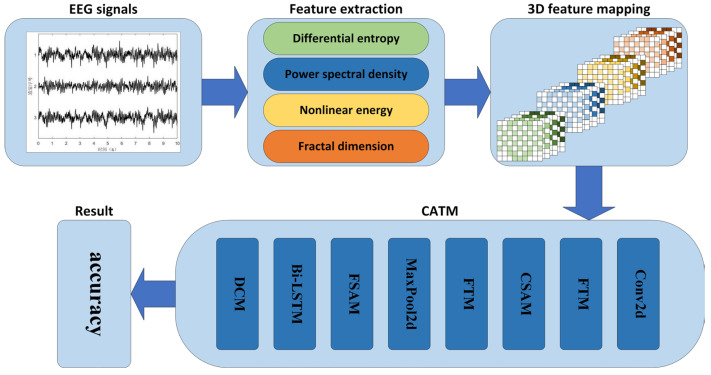
Overall framework and process of the proposed method.

**Figure 2 sensors-24-04837-f002:**
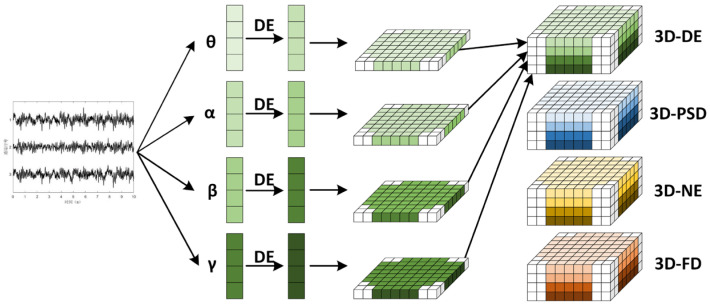
Feature extraction and feature mapping.

**Figure 3 sensors-24-04837-f003:**
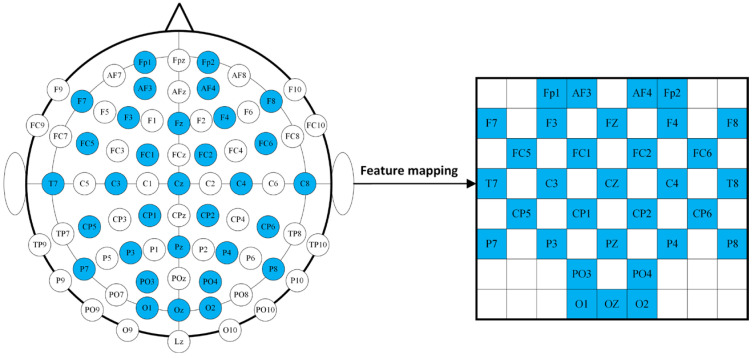
Two-dimensional matrix mapping.

**Figure 4 sensors-24-04837-f004:**
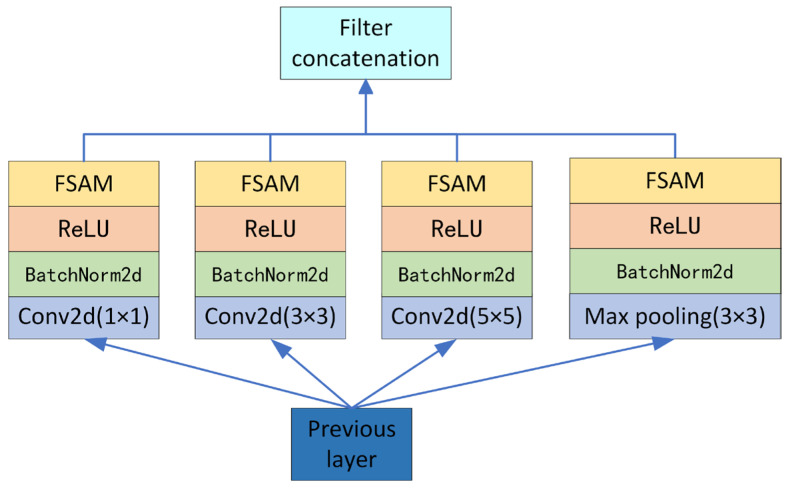
Structural diagram of CSAM.

**Figure 5 sensors-24-04837-f005:**
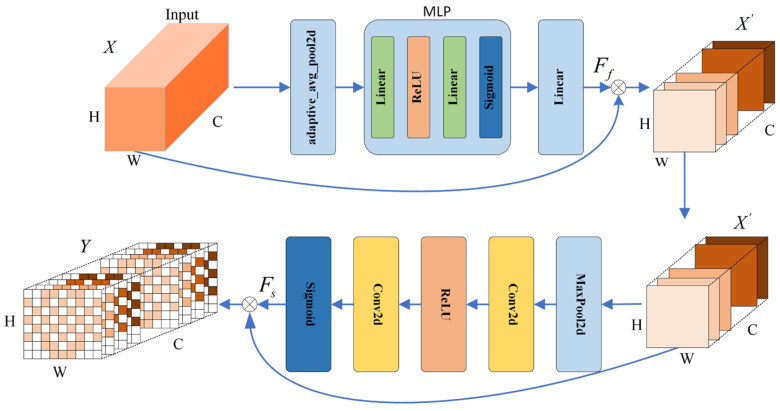
Structural diagram of FSAM.

**Figure 6 sensors-24-04837-f006:**
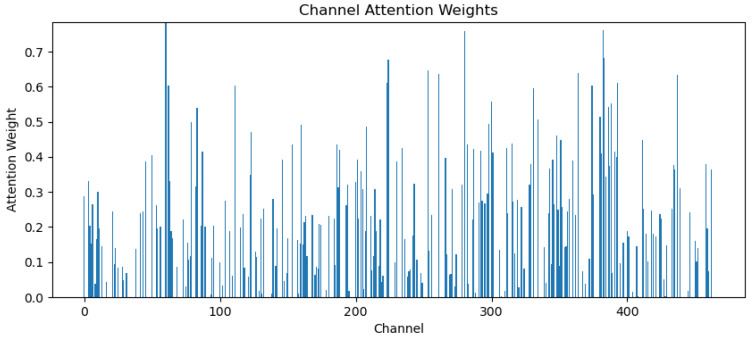
Weights for various channels in CSAM.

**Figure 7 sensors-24-04837-f007:**
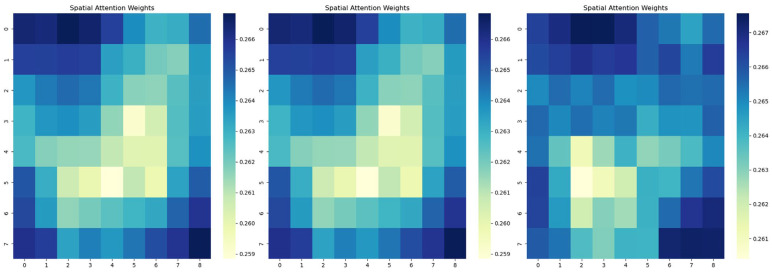
Heat map for each electrode position weight in CSAM.

**Figure 8 sensors-24-04837-f008:**
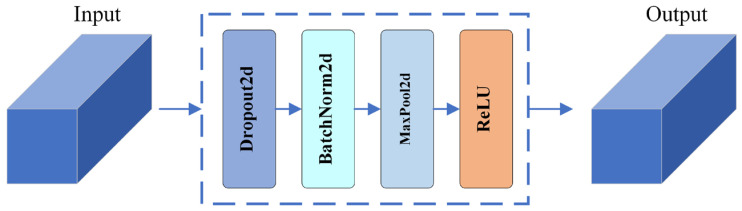
Structural diagram of FTM.

**Figure 9 sensors-24-04837-f009:**
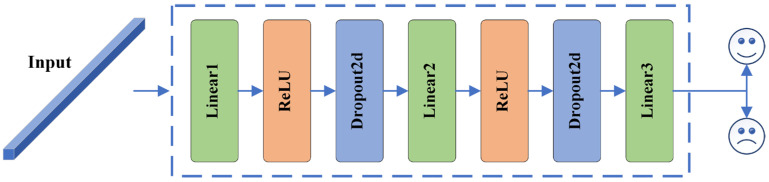
Structural diagram of DCM.

**Figure 10 sensors-24-04837-f010:**
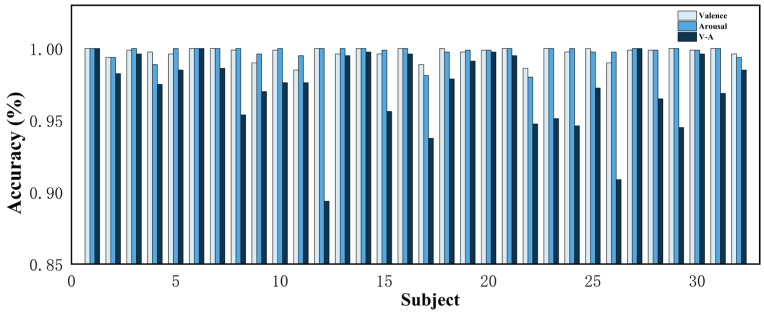
Accuracy of all subjects in the DEAP dataset.

**Figure 11 sensors-24-04837-f011:**
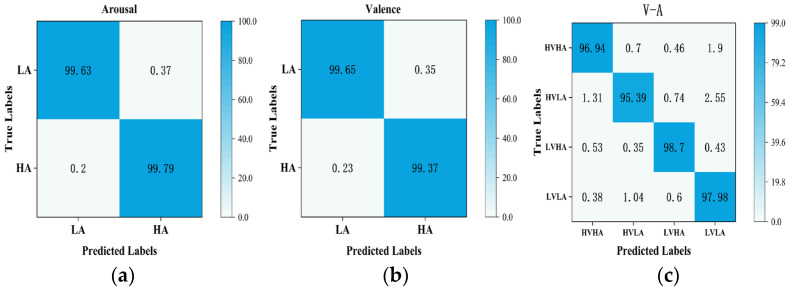
The confusion matrices of the CATM experiment on the DEAP dataset: (**a**) arousal dimension confusion matrix; (**b**) valence dimension confusion matrix; (**c**) valence–arousal dimension confusion matrix.

**Figure 12 sensors-24-04837-f012:**
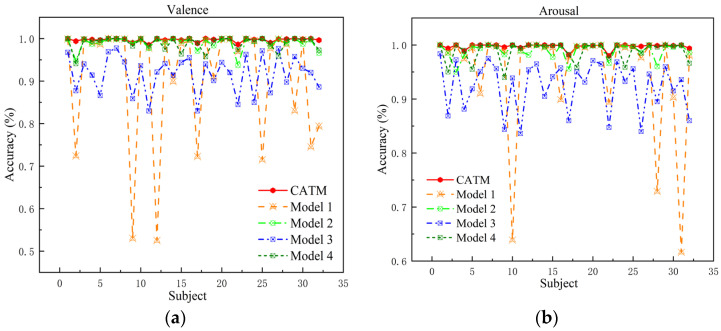
Accuracies of all subjects in different models: (**a**) valence dimension accuracy of all subjects in different models; (**b**) arousal dimension accuracy of all subjects in different models.

**Figure 13 sensors-24-04837-f013:**
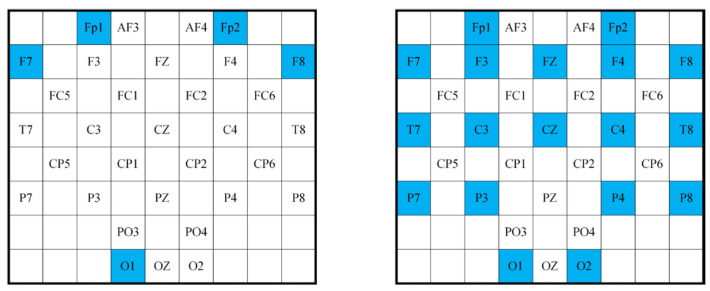
Electrode map with few channels.

**Figure 14 sensors-24-04837-f014:**
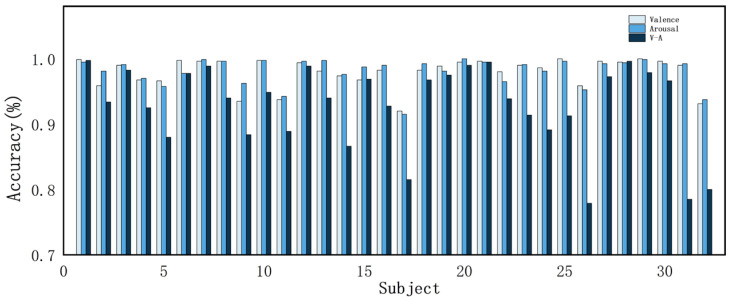
Accuracy of the DEAP dataset across all subjects in the 5-channel experiment.

**Figure 15 sensors-24-04837-f015:**
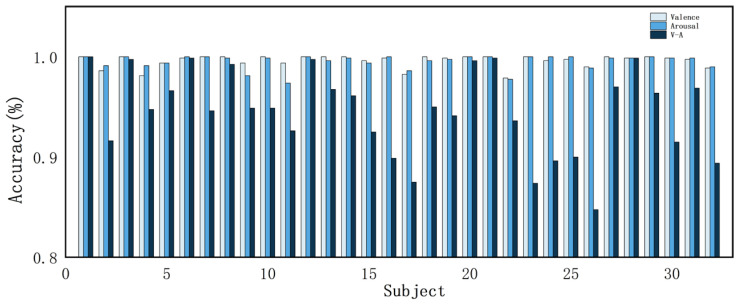
Accuracy of the DEAP dataset across all subjects in the 18-channel experiment.

**Table 1 sensors-24-04837-t001:** Acronyms, full names, and functions of each module of CATM.

Module	Full Name	Function
CSAM	Cross-scale attention module	Extracts features of different scales and assigns weights
FSAM	Frequency–space attention module	Gives higher weight to more important frequency bands and spatial locations
Bi_LSTM	Bidirectional long short-term memory	Extracts time features
DCM	Deep classification module	Classifies the features

**Table 2 sensors-24-04837-t002:** Detailed parameters of CSAM.

Layer	Layer Setting	Output
Conv2D (1 × 1)	In_features = 80	(128 × 128 × 8 × 9)
	Out_feaures = 128	
	BatchNorm2d	
	Activation = ReLU	
Conv2D (3 × 3)	In_features = 80	(128 × 128 × 8 × 9)
	Out_feaures = 128	
	BatchNorm2d	
	Activation = ReLU	
Conv2D (5 × 5)	In_features = 80	(128 × 128 × 8 × 9)
	Out_feaures = 128	
	BatchNorm2d	
	Activation = ReLU	
Max pooling(3 × 3)	In_features = 80	(128 × 80 × 8 × 9)
	Out_feaures = 80	
	BatchNorm2d	
	Activation = ReLU	
Concatenate		(128 × 464 × 8 × 9)

**Table 3 sensors-24-04837-t003:** Detailed parameters of DCM.

Layer	Layer Setting	Output
Linear1	In_features = 128	(128 × 64)
	Out_feaures = 64	
	Activation = ReLU	
Dropout	*p* = 0.5	
Linear2	In_features = 64	(128 × 32)
	Out_feaures = 32	
	Activation = ReLU	
Dropout	*p* = 0.5	
Linear3	In_features = 32 Out_feaures = num_classes	(128 × num_classes)

**Table 4 sensors-24-04837-t004:** Experimental results of different feature fusion methods.

Feature	Accuracy%		Precision%		Recall%		F1-Score%	
	Valence	Arousal	Valence	Arousal	Valence	Arousal	Valence	Arousal
$X_{add}$	98.18	98.59	98.23	98.62	98.18	98.59	98.18	98.59
$X_{mult}$	89.84	88.79	90.08	89.16	89.84	88.79	89.78	88.40
$X_{con}$	99.70	99.74	99.70	99.74	99.69	99.73	99.69	99.73

**Table 5 sensors-24-04837-t005:** Experimental results with different features on the DEAP dataset.

Feature	Accuracy%		Precision%		Recall%		F1-Score%	
	Valence	Arousal	Valence	Arousal	Valence	Arousal	Valence	Arousal
DE	97.39	97.75	98.18	98.28	97.25	97.83	97.6	97.99
PSD	96.44	96.58	97.80	98.34	96.68	97.11	97.06	97.50
NE	95.32	95.51	97.09	97.57	94.59	96.21	95.37	96.66
FD	90.97	97.73	95.10	96.01	90.96	92.93	91.13	92.94
DE-PSD	98.98	98.95	99.25	99.27	98.89	98.99	99.01	99.08
DE-NE	98.81	98.72	99.30	98.85	98.89	98.52	99.03	98.63
DE-FD	98.64	98.88	98.99	99.15	98.50	98.79	98.67	98.92
PSD-NE	97.84	98.33	98.76	98.94	97.81	98.46	98.13	98.64
PSD-FD	98.43	98.40	99.01	99.07	98.38	98.59	98.56	98.76
NE-FD	98.29	98.19	98.59	98.95	97.85	97.72	98.12	97.88
DE-PSD-NE	99.01	99.16	99.48	99.49	99.32	99.16	99.40	99.28
PSD-NE-FD	98.81	99.13	99.22	99.43	98.42	99.16	98.51	99.25
DE-PSD-FD	99.05	99.25	99.49	99.61	99.16	99.36	99.29	99.46
DE-NE-FD	99.14	99.19	99.42	99.03	99.24	99.04	99.33	99.13
All Feature	99.70	99.74	99.70	99.74	99.69	99.73	99.69	99.73

**Table 6 sensors-24-04837-t006:** Ablation experiment models.

Models	CSAM	FSAM	Bi-LSTM	DCM
Model 1	×	√	√	√
Model 2	√	×	√	√
Model 3	√	√	×	√
Model 4	√	√	√	×

The “×” in Table 6 represents a model that does not contain the element on the left, while the “√” represents a model that contains the element. For example, Model 1 is a model that does not contain the CSAM module.

**Table 7 sensors-24-04837-t007:** Ablation results of different modules.

Models	Accuracy/%		Precision/%		Recall/%		F1-Score/%	
	Valence	Arousal	Valence	Arousal	Valence	Arousal	Valence	Arousal
Model 1	91.34	95.06	97.33	98.57	88.61	94.73	87.2	94.15
Model 2	98.95	98.97	99.36	99.27	98.95	98.98	98.95	98.97
Model 3	91.75	92.45	95.07	95.32	91.45	92.62	92.69	93.52
Model 4	98.92	98.87	98.93	98.88	98.92	98.87	98.92	98.87
CATM	99.70	99.74	99.70	99.74	99.69	99.73	99.69	99.73

**Table 8 sensors-24-04837-t008:** Impact of replacing CSAM with single-scale convolution on model performance.

Kernel Size	Accuracy%		Precision%		Recall%		F1-Score%	
	Valence	Arousal	Valence	Arousal	Valence	Arousal	Valence	Arousal
1	88.25	93.73	93.47	96.30	88.25	93.73	84.44	91.77
3	95.12	97.22	97.05	98.24	95.12	97.22	93.66	96.42
5	97.63	98.52	98.50	98.75	97.63	98.52	96.98	98.32

**Table 9 sensors-24-04837-t009:** Results of 5 channels on the DEAP dataset.

Dimension	Accuracy/%	Precision/%	Recall/%	F1-Score/%
Valence	97.96	98.01	97.96	97.95
Arousal	98.11	98.17	98.11	98.10
V-A	92.86	94.12	92.86	91.95

**Table 10 sensors-24-04837-t010:** Results of 18 channels on the DEAP dataset.

Dimension	Accuracy/%	Precision/%	Recall/%	F1-Score/%
Valence	99.59	99.61	99.59	99.59
Arousal	99.53	99.54	99.53	99.53
V-A	94.57	95.98	94.57	93.50

**Table 11 sensors-24-04837-t011:** Recent performance comparison of different methods.

Model	Feature	Dataset	Valence	Arousal	V-A	Year
FSA-3D-CNN	DE	DEAP	95.87%	95.23%	94.53	2022
TSFFN	Baseline removal	DEAP	98.27%	98.53%	-	2022
Multi-aCRNN	DE	DEAP	96.30%	96.43%	-	2023
RA2-3DCNN	Baseline removal	DEAP	97.58%	97.19%	-	2022
MDCNAResnet	DE	DEAP	98.63%	98.89%	-	2023
BiTCAN	Baseline removal	DEAP	98.46%	97.65%	-	2023
RFPN–S2D–CNN	DE	DEAP	96.89%	96.82%	93.56%	2023
FCAN–XGBoost	DE, PSD	DEAP	-	-	95.26%	2023
Multi-scale 3D-CRU	DE	DEAP	93.12%	94.31%	-	2024
MES-CTNet	DE, PSD, SE	DEAP	98.31%	98.28%	-	2024
Ours	DE, PSD, NE, FD	DEAP	99.70%	99.74%	97.27%	2024

## Data Availability

The data for this study were obtained from publicly available datasets. The DEAP dataset is available at http://www.eecs.qmul.ac.uk/mmv/datasets/deap/index.html (accessed on 2 September 2023).

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
