# Peer review of "CATM: A Multi-Feature-Based Cross-Scale Attentional Convolutional EEG Emotion Recognition Model"

_sensors, 2024, doi:10.3390/s24154837_

Round 1

Reviewer 1 Report

Comments and Suggestions for Authors

This study introduces a robust multi-feature, multi-frequency band-based cross-scale attention convolutional model for EEG-based emotion recognition. It significantly improves accuracy by incorporating comprehensive spatial, temporal, and frequency domain data from EEG signals. Evaluated on the DEAP dataset, the proposed approach achieves up to 99.70% and 99.74% accuracy in binary classification tasks, and demonstrates superior performance even with reduced channel setups. The model integrates several innovative components, including cross-scale and frequency-space attention modules, enhancing its sensitivity and precision in feature extraction and classification. Followings are my concerns:

1. Some components of the model, especially the feature transition and deep classification modules, are described but lack detailed justification for their specific configurations. Clarifying these choices could enhance the methodological rigor of the paper.

2. It would be helpful to include more comprehensive details about the dataset partitioning between training and testing phases, as well as any data augmentation techniques employed.

3. More related works could be reviewed such as "EEG-based emotion recognition using hybrid CNN and LSTM classification".

4. How does the CATM model ensure the generalizability of its findings to other EEG datasets beyond DEAP?

5. How does the cross-scale attention module influence the model's performance compared to traditional single-scale models?

6. What are the computational demands of the CATM model, particularly in terms of processing time and hardware requirements?

7. Please fix the misprints in the manuscripts such as the double numbering of the equations.

Author Response

请参阅附件。

Reviewer 2 Report

Comments and Suggestions for Authors

The article presents CATM, a cross-scale attentional convolutional model that integrates multiple features from EEG signals across time, frequency, and spatial domains to improve the accuracy of emotion classification. The architecture includes a cross-scale attention module for spatial feature extraction at varying scales, a frequency-space attention module that prioritizes significant channels and spatial areas, a temporal feature extraction module, and a depth classification module for final emotion categorization. Evaluated on the DEAP dataset, the model achieved binary classification accuracy rates of 99.70% and 99.74% for valence and arousal, respectively, and 97.27% for a four-class valence-arousal classification. In reduced-channel scenarios, using only five channels, the model still delivered high accuracy levels of 97.96% and 98.11% for valence and arousal binary classifications and 92.86% for the four-class scenario. These results highlight the effectiveness of the proposed model in utilizing multi-feature and multi-frequency band information for improved emotion recognition from EEG signals. The objective of this work is clear, and the proposed technique sounds available. However, several vital issues must be improved, as listed below:

1. This manuscript exhibits many acronyms, so it would be better to include a table that clarifies the acronyms employed, provides their corresponding full names, and describes their usage.

2. In Section 2, I suggest that the author first introduce the brain rhythms in the EEG and then concentrate on related works.

3. Why only four brain rhythms have been used in this work? I have found delta waves can also be applied in previous EEG-based emotion classification works.

4. The related work should include the previous channel selection techniques in the EEG studies, as this study also focuses on a few channel experiments.

5. As for the results, it is necessary to combine neuroscience and psychology to explain the rationality of the selected channels.

6. To prove the selected number, I suggest the authors discuss the relationship between the number of EEG channels applied and the performance based on the proposed method.

7. Following the above two concerns, please provide additional insights into the results obtained from the reduced channel experiments. Elaborate on how the deep learning model performs with fewer channels and discuss potential applications where a limited number of channels could be advantageous.

8. The authors should provide details concerning the overall complexity and time consumption (or running time) of the proposed network model. To make this work more impressive, I suggest comparing such factors to the other models.

9. There are various shapes of window functions, and which one has been used in the sliding window? Besides, is a 0.5 s time window suitable for feature extraction? 

10. It would also be very desirable if a better and clearer graphical presentation could be used in the revised version. The current figures are low quality.

Comments on the Quality of English Language

The authors should spend time revising this manuscript. The technical writing requires further improvement, as there are many grammatical errors, typos, and improper wording. Moreover, the flow of this work also needs to be improved. For example, Section 3.1 and Section 4.1 should be combined into a Section called “Experiment”. I suggest the authors review and follow the high-quality papers in this field and then reorganize this work.   

Round 2

Reviewer 2 Report

Comments and Suggestions for Authors

The authors addressed my queries, and I am satisfied with the answers. The manuscript has been improved, so I recommend it for acceptance.